# Loneliness and Increased Hazardous Alcohol Use: Data from a Nationwide Internet Survey with 1-Year Follow-Up

**DOI:** 10.3390/ijerph191912086

**Published:** 2022-09-24

**Authors:** Mami Wakabayashi, Yoshifumi Sugiyama, Midori Takada, Aya Kinjo, Hiroyasu Iso, Takahiro Tabuchi

**Affiliations:** 1Institute for Global Health Policy Research, Bureau of International Health Cooperation, National Center for Global Health and Medicine, 1-21-1, Toyama Shinjuku-ku, Tokyo 162-8655, Japan; 2Division of Clinical Epidemiology, Research Center for Medical Sciences, The Jikei University School of Medicine, Tokyo 105-8461, Japan; 3Department of Cardiovascular Disease Prevention, Osaka Center for Cancer and Cardiovascular Disease Prevention, Osaka 536-8588, Japan; 4Division of Environmental and Preventive Medicine, Department of Social Medicine, Faculty of Medicine, Tottori University, Tottori 683-8503, Japan; 5Cancer Control Center, Osaka International Cancer Institute, Osaka 541-8567, Japan; 6The Tokyo Foundation for Policy Research, Tokyo 106-6234, Japan

**Keywords:** alcohol use disorders identification test, hazardous alcohol use, alcohol use disorders, loneliness, COVID-19

## Abstract

We aimed to examine the association between loneliness and developing alcohol dependence or hazardous alcohol use. A cohort study was conducted utilizing data from a nationwide internet survey in 2021 and 2022 in Japan. A total of 15,854 follow-up participants (55% men, with a mean age of 52.8 years) were divided based on AUDIT scores: nondrinkers (AUDIT: 0), low-risk drinkers (AUDIT: 1–7), medium-risk drinkers (AUD: 8–14), high-risk drinkers (AUDIT: 15–19), and probable alcohol dependence (AUDIT: 20–40). The University of California, Los Angeles Loneliness Scale (Version 3), a short-form three-item scale, was used to assess loneliness (high loneliness score of ≥6). The prevalence of high loneliness was higher in nondrinkers than that in low- and medium-risk drinkers, i.e., 22%, 18%, and 17%, respectively, as well as in high-risk drinkers (32%) and those with probable alcohol dependence (43%) compared to non-high-risk drinkers (19%). After adjusting for various factors (sociodemographic, social isolation, psychological distress, and smoking), non-high-risk drinkers (AUDIT: 0–14) with high loneliness were more likely to become high-or-over-risk drinkers (AUDIT: 15–40) than those without high loneliness, with adjusted risk ratios of 1.45 (95% confidence interval: 1.08–1.96) through multivariable binary logistic regression. Among non-high-risk drinkers, people with high loneliness scores at baseline were associated with increased high-risk drinking patterns with probable alcohol dependence.

## 1. Introduction

Mental disorders are associated with poor health behaviors; mentally distressed individuals are more likely to engage in unhealthy behaviors such as harmful alcohol use [1]. Various studies reported that lonely people who engaged in heavy drinking were more vulnerable to alcohol-related problems due to a lack of social support as well as community peer pressure [2,3,4]. However, there is no clear evidence on whether loneliness is likely to develop alcohol dependence or hazardous alcohol use [1,5,6]. Thus, identifying causal correlations between loneliness and hazardous alcohol use could have high public health relevance.

Alcohol use is a major global concern associated with disease onset and often leads to death [7,8]. Quarantine and social isolation during the COVID-19 pandemic have resulted in increased alcohol consumption and alcohol abuse [9,10,11]. The government of Japan has implemented preventive measures, such as quarantine, gathering regulations, reducing business hours for restaurants, and regulation of serving alcoholic beverages at restaurants [12]. The regulations for the prevention of COVID-19 were enforced for almost 7 months in 2021 and from January to mid-March of 2022 (as of September 2022) at prefectures under a “State of Emergency” in Japan [13]. As such, it is plausible to concur that people have experienced loneliness throughout the pandemic [14]. This study aimed to examine the association between loneliness and developing alcohol dependence or hazardous alcohol use during the COVID-19 pandemic.

## 2. Materials and Methods

### 2.1. Data Source and Study Population

#### 2.1.1. Internet Survey

We used data from the Japan Society and New Tobacco Internet Survey (JASTIS), a large internet-based cohort study that has focused on tobacco issues since 2015 and has expanded to various health behaviors since 2021. The JASTIS cohort profile has been previously published [15]. The present study utilized two waves of longitudinal data from February 2021 (baseline, JASTIS2021) to February 2022 (JASTIS2022) and examined how the COVID-19 pandemic affected individuals’ health behaviors in Japan. This web-based, self-reported questionnaire survey was administrated by a large internet research agency, Rakuten Insight, Inc., which pooled approximately 2.3 million panelists. Survey requests were sent by the research agency to the panelists, who were each selected based on sex, age, and prefecture. The panelists who consented to participate accessed the designated website and responded to the survey. The participants were given the option not to respond to any part of the survey or discontinue it at any point. The survey was closed when the target number of respondents for sex, age, and prefecture was met. The total number of participants in each survey was 26,000 and 33,000 participants in 2021 and 2022, respectively.

#### 2.1.2. Managing the Data Quality and Generating the Study Population

To ensure data quality, respondents with discrepancies or artificial/unnatural responses were excluded from the study [15]. In this regard, the following three items were used: (1) “Please choose the second from the bottom”, (2) “choosing ‘yes’ in all questions on alcohol and nine illegal drug uses”, and (3) “choosing ‘yes’ in all questions regarding nine chronic diseases”. Moreover, 871 respondents who had inconsistent answers to alcohol-related questions were excluded: respondents who answered “Never” in the question “Do you drink alcohol?” but answered “monthly or more” in the question “How often do you have a drink containing alcohol?”. In addition, the study participants were limited only to individuals aged ≥20 years, the legal drinking age in Japan. Overall, the total number of follow-up participants was 15,854. The selection flow of the study population is shown in Appendix A.

### 2.2. Measures

#### 2.2.1. Explanatory Variables

A Japanese version of the University of California, Los Angeles Loneliness Scale (Version 3), a short-form three-item scale (UCLA-LS-SF3), was used to assess loneliness. UCLA-LS-SF3 was initially developed in English [16], and its Japanese version was previously validated and found to be reliable [17,18]. Participants were asked the following three questions about their condition over the past 30 days: “How often do you feel that you lack companionship?”, “How often do you feel left out?”, and “How often do you feel isolated from others?”. Their responses to these questions were recorded using a 5-point scale used in JASTIS2021 (1: never, 2: rarely, 3: sometimes, 4: usually, and 5: always). We integrated these recorded scales into a 3-point scale (1 hardly ever [never and rarely], 2 some of the time [sometime], and 3 often [usually and always]) based on UCLA-LS-SF3. The total score range of these three questions was 3 to 9, categorized into two groups: people with a score of 3–5 were designated as “non-low loneliness” and people with a score of 6–9 were designated as “high loneliness” [16,19]. The Cronbach’s alpha value of the internal consistency for the three items was 0.93.

#### 2.2.2. Outcome Variables

Nondrinkers and current drinkers were identified by the AUDIT, which was developed by the WHO and is one of the most effective screening tools in identifying individuals with alcohol use disorders and those involved in harmful alcohol use [20]. The AUDIT is a 10-item screening measure that assesses alcohol use during the last 12 months [21,22]. Although the AUDIT does not intend to offer a clinical diagnosis, it can indicate the presence and severity of alcohol problems or alcohol use disorders [23]. Furthermore, the AUDIT translated into Japanese was validated to identify harmful to hazardous alcohol use and alcohol use disorders as a screening tool [24]. Each item in the AUDIT has a score of 0–4 points, except items 9 and 10, which inquire about alcohol-related injury or violence, with scores of 0, 2, or 4 points (the details of each question are shown in Appendix A). The total scores range from 0 to 40 points. According to the National Health Guidance and Brief Intervention to Support Alcohol Reduction in Japan based on a validation study, an AUDIT score of 0–7 is considered normal, people with a score 8–14 are advised to reduce hazardous drinking, and people with a score ≥ 15 are recommended to see a specialist for further diagnostic evaluation of alcohol dependence [25]. On the other hand, the AUDIT original guideline proposed by the WHO interprets the AUDIT scores as follows: people with a score of 8–15 are advised to focus on the reduction of hazardous drinking, people with a score of 16–19 are advised to receive brief counseling and be monitored continuously, and people with a score ≥ 20 clearly warrant further diagnostic evaluation for alcohol dependence [21]. Considering these two guidelines, the participants of this study were categorized into five groups based on the total score: nondrinker (AUDIT: 0), low-risk drinker (AUDIT: 1–7), medium-risk drinker (AUDIT: 8–14), high-risk drinker (AUDIT: 15–19), and probable alcohol dependence (AUDIT: 20–40) [21,25,26]. Participants who were categorized as having high-risk alcohol use and probable alcohol dependence were identified as high-or-over-risk drinkers (AUDIT: 15–40) for the cutoff point in alcohol pattern changes regarding an association of the prevalence of loneliness, whereas other participants were regarded as non-high-risk drinkers (AUDIT: 0–14), including nondrinkers.

Changes in AUDIT scores from 2021 to 2022 were categorized into four patterns as follows: no changes from non-high-risk drinkers (0–14) to non-high-risk drinkers (0–14), non-high-risk drinkers (0–14) to high-or-over-risk drinkers (15–40), no changes from high-or-over-risk drinkers (15–40) to high-or-over-risk drinkers (15–40), and high-or-over-risk drinkers (15–40) to non-high-risk drinkers (0–14).

#### 2.2.3. Demographics and Potential Health Factors Related to Alcohol Use

Demographic questions inquired about the following variables: age, sex, educational level, job, equivalent annual household income, marital status, and current living arrangements. Educational level was categorized as low (graduated from high school or lower), middle (graduated from vocational or junior college), and high (graduated from university or higher). Equivalent annual household income is the household income divided by the square root of the number of household members. This factor is categorized as <2 million yen, 2–4 million yen, 4–6 million yen, 6–10 million yen, ≥10 million yen, and do not know/want to answer. Jobs were categorized into regular jobs including self-employment, non-regular employee, no main income job (students, retirees, and housework only), and unemployed. Marital status was categorized as married, single, and divorced/widowed. The current living arrangements reported whether the participants lived with someone or alone. Finally, potential factors related to alcohol use include current smoker and the Kessler Psychological Distress Scale (K6: ≥13) [27].

### 2.3. Statistical Analyses

Statistical analyses were performed using the Stata MP version 15 (StataCorp LLC, College Station, TX, USA). The means and standard deviations were presented as continuous variables and proportions for categorical variables. First, we determined variations in the means and proportions of demographics, and the potential health factors based on AUDIT scores for each category. The difference (*p* for difference) between non-high-risk drinkers and high-or-over-risk drinkers was assessed using the t-test for the mean and using the chi-square test for the proportion. Second, multivariable binary logistic regression was performed among non-high-risk drinkers at baseline to examine the relationship between loneliness and alcohol use changes from non-high-risk drinkers to high-or-over-risk drinkers. Model 1 was univariable, Model 2 was adjusted for sociodemographic factors, and Model 3 was further adjusted for marital status and living arrangement as potential social isolation factors [1] and smoking. Model 4 was further adjusted for K6 as psychological distress due to strong correlations with loneliness and hazardous alcohol use under the COVID-19 preventive measures. Finally, the adjusted risk ratios (aRR) and 95% confidence intervals (CIs) for AUDIT pattern changes were observed. All statistical tests were two-sided, and a *p* value < 0.05 was considered statistically significant.

### 2.4. Ethical Approval

All procedures were conducted as per the ethical standards of the 1975 Helsinki Declaration (revised in 2013). The Research Ethics Committee of the Osaka International Cancer Institute reviewed and approved the study protocol (approval No. 1412175183). All participants provided their informed consent before responding to the online questionnaire. Furthermore, the internet survey agency respected the Act on Protection of Personal Information in Japan. As an incentive, credit points (known as “E-points”), which can be used for internet shopping and cash conversion, were provided to participants.

## 3. Results

### 3.1. Participant Characteristics

Of the 15,854 follow-up participants, 52% (n = 8239) were men with a mean age of 52.8 (±15.6) (range, 20–80) years at baseline. Based on the participant characteristics, sociodemographic categories with the highest proportion of responses were a high education level (48%), married (64%), living with someone (81%), having a regular job (43%), and income level of 2–4 million (36%) (Table 1). This study also included current smokers (20%) and those with a K6 score of ≥13 (10%). The distribution of each characteristic of the follow-up population was similar to that of participants from JASTIS2021 and JASTIS2022, although participants who could not be followed up in 2022 were younger, more likely females, and had a lower percentage of K6 scale score than those who were followed up (data not shown).

### 3.2. Characteristics of Participants according to AUDIT Scores

The number of high-or-over-risk drinkers (AUDIT ≥ 15) was 763 (5% of the follow-up population) (Table 1). In addition, their mean age was 49.4 (±13.7) years. Most of the high-or-over-risk drinkers were aged 40–59 years (51%), males (78%), had a high education level (54%), were living alone (23%), had a regular job (68%), earned 4–6 million yen (19%), current smokers (40%), and had a K6 score of ≥13 (21%).

### 3.3. Loneliness and Hazardous Alcohol Use Based on AUDIT Scores

The mean score for loneliness was 4.1 (±1.8), and the proportion of participants with high loneliness score (≥6) was 20% among the followed-up participants. The proportion of participants with high loneliness scores in nondrinkers was 22%, which was higher than that in low-risk (18%) and medium-risk drinkers (17%). The proportion of high-risk drinkers with high loneliness scores was 32%, whereas the proportion of those with probable alcohol dependence having high loneliness scores was 43%. The proportion of high-or-over-risk drinkers with high loneliness scores was twice as high as the proportion of non-high-risk drinkers with high loneliness scores. Statistical analysis revealed that the means of loneliness score were significantly different between non-high-risk drinkers and high-or-over-risk drinkers (*p* < 0.05).

### 3.4. Association between Loneliness and AUDIT Score Pattern Changes

Table 2 shows the mean and proportion of loneliness scores based on AUDIT score pattern changes from 2021 to 2022. A greater proportion of participants with high loneliness scores was observed in the pattern of non-high-risk drinkers (AUDIT: 0–14) to high-or-over-risk drinkers (AUDIT: 15–40) compared with those in the non-high-risk drinkers (AUDIT: 0–14) to non-high-risk drinkers (AUDIT: 0–14) (respective proportions were 27% and 19%). Furthermore, the proportion of individuals with high loneliness scores in the no-pattern changes from high-or-over-risk drinkers to high-or-over-risk drinkers had a slightly higher proportion of high loneliness scores than those in the pattern changes from high-or-over to non-high-risk drinkers (39% vs. 36%; Appendix A).

Table 3 presents the aRR for the AUDIT score pattern changes from non-high-risk to high-risk drinkers related to loneliness among non-high-risk drinkers at baseline. In the univariable model (Model 1), participants with high loneliness scores at baseline had an association with the changing patterns from non-high-risk to high-risk drinkers compared with participants with non-low loneliness (RR: 1.56, 95% CI: 1.24–1.97). The association was a little attenuated after adjusting for sociodemographic factors, including sex, in Model 1. A slight change in the association between loneliness and changes in high-risk drinking patterns was seen after adjusting for potential social isolation factors (marital status and living arrangements) and smoking in Model 3. After further adjustment for psychological distress in Model 4, participants with high loneliness scores at baseline had an association with the changing patterns from non-high-risk to high-risk drinkers compared with participants with non-low loneliness (aRR: 1.45, 95% CI: 1.08–1.96).

## 4. Discussion

In this nationwide longitudinal study, loneliness was associated with increased severity of hazardous alcohol drinking patterns with 1-year follow-up during the COVID-19 pandemic, independent of objective social isolation, psychological distress, and other potential confounders. We found that nondrinkers scored slightly higher on loneliness than low- and medium-risk drinkers. Nondrinkers had much lower loneliness scores than high-risk drinkers and those with probable alcohol dependence.

Drinking is a social facilitator. Previous studies showed that elderly people enjoyed alcoholic beverages during social gatherings in Australia [28], and social networks from relatives and friends were associated with individuals’ drinking behaviors in the U.S. [29]. During the COVID-19 pandemic, social gatherings, such as dining out with a large group, were regulated by governments as a preventive measure against transmission worldwide. This led to social isolation, and in turn, increased feelings of loneliness [30]. In Japan, the prevalence of social isolation increased by 6.7% during the COVID-19 pandemic [31]. However, only a few studies have examined the impact of social isolation and loneliness on increased risky drinking patterns. Our results suggested that social isolation had a comparable impact on increased risky drinking patterns as well as loneliness, probably because living alone during the COVID-19 pandemic led to social isolation due to the regulation of gatherings, restriction of dining out with groups, and staying at home.

Loneliness has been reported to be associated with mental disorders, such as depressive symptoms [32]. Psychological distress due to COVID-19 was associated with higher rates of high-risk drinking [33,34]. A possible causal relationship may occur between psychological distress and increased high-risk drinking in a public health emergency situation. However, our results showed that loneliness was an independent risk factor for increased risk of drinking patterns after adjusting for psychological distress.

In a previous cross-sectional study, loneliness was not associated with at-risk drinking or binge drinking in the U.S. [6]. In the abovementioned previous study, at-risk drinking was defined as the volume of average drinking per week and per day, and nondrinkers were defined as having no alcohol consumption in the last 3 months. Another study showed no clear evidence between loneliness and hazardous alcohol use or alcohol dependence (based on genetic variants) [5]. On the other hand, a cross-sectional survey of the nine countries of the former Soviet Union showed that loneliness was associated with alcohol consumption, hazardous drinking, and problem drinking, although no association was observed in some of the countries [35]. We noted that the relationship between loneliness and drinking was not linear; low- and medium-risk drinkers may feel less lonely than nondrinkers, possibly owing to social interaction during drinking. In our study, we did not distinguish between lifelong abstinence, current abstainers, and former problem drinkers among nondrinkers. Previous studies showed that current abstainers and former problem drinkers did not drink probably due to their health status [36,37] and former drinkers were less healthy than current drinkers and lifelong abstainers [38]. In addition, nondrinkers tended to have worse socioeconomic status compared to the general population and current drinkers [37]. The definition of nondrinkers and/or at-risk drinkers may lead to inconsistent results in previous cross-sectional studies.

Our results suggested that loneliness triggered the development of hazardous alcohol use as defined by AUDIT. Loneliness scores among drinkers with symptoms such as lack of control when drinking; feeling guilty or experiencing blackouts after drinking; and alcohol-related injuries, including hurting others, were higher than those in drinkers who did not present these symptoms (Appendix A). The effective interventions for reducing loneliness were explored, such as meditation/mindfulness, social cognitive training, and social support [39]. Those interventions may be helpful for people with high loneliness to prevent hazardous alcohol use.

The main strength of this study was its large sample size due to the nationwide sampling design to evaluate the relationship between loneliness and the development of hazardous alcohol use, including probable alcohol dependence. However, this study also had several limitations. First, we used self-reported loneliness measures and drinking patterns based on AUDIT, which could have led to measurement errors. Nonetheless, both self-reported loneliness measures and drinking patterns based on AUDIT are accepted as relevant screening tests and are reliable and valid in Japanese individuals [24,40]. Second, the current study design restricted the results from being generalized to the general Japanese populations. Our data showed that 5% of people (7% for men and 2% for women) had an AUDIT score ≥15, which was greater than that reported in the national survey (2.9%; 5.2% for men and 2.9% for women) in 2018 [41]. However, an increase in hazardous alcohol use during the COVID-19 pandemic was reported in the U.S. [42,43], suggesting that the prevalence of high-risk alcohol drinkers being higher in our study than that reported in the 2018 national survey was valid. Third, our follow-up population with data on all variables slightly differed from the overall sample, which could have led to a selection bias. For example, the follow-up population was older than the overall population. Yet, similar prevalence of main variables, including loneliness and AUDIT, between the follow-up population and overall population allowed us to examine the association between loneliness and the development of high-risk drinking. Fourth, we were unable to compare the current data with those reported in the pre-pandemic period. The preventive measures aimed at reducing COVID-19 transmission might have affected the survey outcomes. Further investigations are warranted to clarify this under no social restrictions. Fifth, although we adjusted for a wide range of confounders, the possibility of residual confounding, such as duration of high loneliness and drinking styles (e.g., social drinking, drinking at home, and drinking alone), cannot be excluded.

## 5. Conclusions

This study examined the association between loneliness and increased hazardous alcohol consumption during the COVID-19 pandemic. Individuals experiencing high loneliness were more susceptible to engaging in hazardous alcohol use. Providing appropriate support to people experiencing high degrees of loneliness, such as social supports, will have a positive impact on the prevention of potential alcohol dependence.

## Figures and Tables

**Table 1 ijerph-19-12086-t001:** Characteristics of the follow-up participants based on categories, 2021.

	Follow-Up	Drinking Categories Based on AUDIT							
		Non-High-Risk (0–14)	Nondrinker (0)	Low-Risk (1–7)	Medium-Risk (8–14)	High-or-Over-Risk (≧15)	High-Risk (15–19)	Probable Alcohol Dependence (20–40)	
	N = 15,854	N = 15,091 (95.2%)	N = 4719 (29.8%)	N = 8749 (55.2%)	N = 1623 (10.2%)	N = 763 (4.8%)	N = 404 (2.5%)	N = 359 (2.3%)	*p* Value *
Loneliness (Mean, SD)	4.1 (±1.8)	4.0 (±1.8)	4.2 (±1.9)	4.0 (±1.75)	3.9 (±1.7)	4.9 (±2.2)	4.6 (±2.1)	5.1 (±2.3)	<0.001
Score < 6 (%)	12,648 (80)	12,172 (81)	3669 (78)	7157 (82)	1346 (83)	476 (62)	273 (68)	203 (57)	<0.001
Score ≧ 6 (%)	3206 (20)	2919 (19)	1050 (22)	1592 (18)	277 (17)	287 (38)	131 (32)	156 (43)	
Age, year (Mean, SD)	52.8 (±15.6)	52.9 (±15.7)	52.5 (±16.0)	52.9 (±15.8)	54.6 (±14.0)	49.4 (±13.7)	50.3 (±14.2)	48.4 (±13.0)	<0.001
20–39 years (%)	3513 (22)	3337 (22)	1144 (24)	1942 (22)	251 (15)	176 (23)	96 (24)	80 (22)	<0.001
40–59 years (%)	6246 (39)	5855 (39)	1778 (38)	3381 (39)	696 (43)	391 (51)	189 (47)	202 (56)	
60 or over (%)	6095 (38)	5899 (39)	1797 (38)	3426 (39)	676 (42)	196 (26)	119 (29)	77 (21)	
Sex (%)									<0.001
Women	7615 (48)	7449 (49)	2895 (61)	4209 (48)	345 (21)	166 (22)	69 (17)	97 (27)	
Men	8239 (52)	7642 (51)	1824 (39)	4540 (52)	1278 (79)	597 (78)	335 (83)	262 (73)	
Education (%)									<0.001
Low	4899 (31)	4647 (31)	1704 (36)	2483 (28)	460 (28)	252 (33)	123 (30)	129 (36)	
Middle	3342 (21)	3244 (22)	1168 (25)	1827 (21)	249 (15)	98 (13)	47 (12)	51 (14)	
High	7613 (48)	7200 (48)	1847 (39)	4439 (51)	914 (56)	413 (54)	234 (58)	179 (50)	
Marital status (%)									0.919
Marriage	10,121 (64)	9631 (64)	2792 (59)	5687 (65)	1152 (71)	490 (64)	271 (67)	219 (61)	
No marriage	4085 (26)	3893 (26)	1327 (28)	2237 (26)	329 (20)	192 (25)	96 (24)	96 (27)	
Divorced/Widowed	1648 (10)	1567 (10)	600 (13)	825 (9)	142 (9)	81 (11)	37 (9)	44 (12)	
Living arrangements (%)									0.004
Living with someone	12,849 (81)	12,261 (81)	3788 (80)	7155 (82)	1318 (81)	588 (77)	322 (80)	26,674)	
Living alone	3005 (19)	2830 (19)	931 (20)	1594 (18)	305 (19)	175 (23)	82 (20)	93 (26)	
Job (%)									<0.001
Regular job	6823 (43)	6308 (42)	1573 (33)	3793 (43)	942 (58)	515 (68)	279 (69)	236 (66)	
Non-regular employee	2964 (19)	2856 (19)	953 (20)	1658 (19)	245 (15)	108 (14)	52 (13)	56 (16)	
No main income job as students/retiree/ housework	3798 (24)	3753 (25)	1412 (30)	2146 (25)	195 (12)	45 (6)	23 (6)	22 (6)	
Unemployed	2269 (14)	2174 (14)	781 (17)	1152 (13)	241 (15)	95 (12)	50 (12)	45 (13)	
Income (%)									<0.001
Under 2 million yen	2537 (16)	2417 (16)	960 (20)	1229 (14)	228 (14)	120 (16)	61 (15)	59 (16)	
2–4 million yen	5769 (36)	5491 (36)	1634 (35)	3283 (38)	574 (35)	278 (36)	153 (38)	125 (35)	
4–6 million yen	2550 (16)	2403 (16)	588 (12)	1501 (17)	314 (19)	147 (19)	72 (18)	75 (21)	
6–10 million yen	1650 (10)	1532 (10)	322 (7)	976 (11)	234 (14)	118 (15)	71 (18)	47 (13)	
10 million or more	367 (2)	334 (2)	77 (2)	191 (2)	66 (4)	33 (4)	21 (5)	12 (3)	
Don’t know/want to answer	2981 (19)	2914 (19)	1138 (24)	1569 (18)	207 (13)	67 (9)	26 (6)	41 (11)	
Current smoker (%)									<0.001
No	12,763 (81)	12,305 (82)	4093 (87)	7109 (81)	1103 (68)	458 (60)	237 (59)	221 (62)	
Yes	3091 (20)	2786 (18)	626 (13)	1640 (19)	520 (32)	305 (40)	167 (41)	138 (38)	
Psychological Distress (Mean, SD)	4.6 (±5.6)	4.5 (±5.5)	4.8 (±5.8)	4.3 (±5.3)	4.4 (±5.5)	7.3 (±6.6)	6.6 (±5.9)	8.1 (±7.2)	<0.001
K6 < 13 (%)	14,268 (90)	13,669 (91)	4214 (89)	7992 (91)	1463 (90)	599 (79)	334 (83)	265 (74)	<0.001
K6 ≧ 13 (%)	1586 (10)	1422 (9)	505 (11)	757 (9)	160 (10)	164 (21)	70 (17)	94 (26)	

* The difference in mean and difference in proportion of each category was based on two drinking categories between non-high-risk drinking and high-risk drinking.

**Table 2 ijerph-19-12086-t002:** The change of AUDIT patterns from 2021 to 2022 and associated factors based on 2021 among non-high-risk drinking patterns, 2021.

	Non-High-Risk at Baseline N = 15,091	
	Non-High-Risk to Non-High-Risk (0–14) to (0–14)	Non-High-Risk to High-or-Over (0–14) to (15≧)
	N = 14,742	N = 349
	N (%)	N (%)
Loneliness (Mean, SD)	4.0 (±1.8)	4.4 (±1.9)
Score < 6	11,918 (81)	254 (73)
Score ≧ 6	2824 (19)	95 (27)
Age, year (Mean, SD)	53.0 (±15.7)	49.9 (±14.5)
20–39 years	3246 (22)	91 (26)
40–59 years	5695 (39)	160 (46)
60–79 years	5801 (39)	98 (28)
Sex		
Women	7382 (50)	67 (19)
Men	7360 (50)	282 (81)
Education		
Low	4561 (31)	86 (25)
Middle	3173 (22)	71 (20)
High	7008 (48)	192 (55)
Marital status		
Marriage	9398 (64)	233 (67)
No marriage	3810 (26)	83 (24)
Divorced/Widowed	1534 (10)	33 (9)
Living arrangements		
Living with someone	11,987 (81)	274 (79)
Living alone	2755 (19)	75 (21)
Job		
Regular job	6093 (41)	215 (62)
Non-regular employee	2800 (19)	56 (16)
No main income job, students/retiree/housework	3716 (25)	37 (11)
Unemployed	2133 (14)	41 (12)
Income		
Under 2 million yen	2366 (16)	51 (15)
2–4 million yen	5354 (36)	137 (39)
4–6 million yen	2342 (16)	61 (17)
6–10 million yen	1477 (10)	55 (16)
10 million or more	327 (2)	7 (2)
Don’t know/want to answer	2876 (20)	38 (11)
Current smoker		
No	12,075 (82)	230 (66)
Yes	2667 (18)	119 (34)
Psychological Distress (Mean, SD)	4.4 (±5.4)	6.0 (±6.2)
K6 < 13	13,370 (91)	299 (86)
K6 ≧ 13	1372 (9)	50 (14)

**Table 3 ijerph-19-12086-t003:** Risk ratios of loneliness for increased high-risk drinking by AUDIT pattens from (0–14) in 2021 to (15–40) in 2022 among non-high-risk drinking patterns at baseline (N = 15,901).

	Model 1	Model 2	Model 3	Model 4
	RR (95%CI)	aRR (95%CI)	aRR (95%CI)	aRR (95%CI)
Loneliness				
Score < 6	ref	ref	ref	Ref
Score ≧ 6	**1.56 (1.24–1.97)**	**1.52 (1.18–1.96)**	**1.57 (1.22–2.03)**	**1.45 (1.08–1.96)**
Age				
20–39 years		ref	ref	Ref
40–59 years		0.97 (0.73–1.27)	0.81 (0.61–1.07)	0.81 (0.61–1.08)
60–79 years		0.76 (0.54–1.06)	**0.57 (0.39–0.81)**	**0.58 (0.40–0.83)**
Sex				
Women		ref	ref	ref
Men		**4.43 (3.17–6.19)**	**4.26 (3.01–6.02)**	**4.27 (3.02–6.04)**
Education				
Low		ref	ref	ref
Middle		1.36 (0.97–1.91)	1.39 (0.99–1.94)	1.38 (0.99–1.94)
High		1.05 (0.80–1.38)	1.10 (0.83–1.44)	1.09 (0.83–1.43)
Job				
Regular job		ref	ref	ref
Non-regular employee		1.06 (0.76–1.49)	1.17 (0.83–1.66)	1.18 (0.83–1.66)
No main income job as students/retiree/housework		0.73 (0.47–1.12)	0.81 (0.53–1.26)	0.81 (0.52–1.26)
Unemployed		0.76 (0.49–1.16)	0.87 (0.57–1.34)	0.87 (0.56–1.34)
Income				
Under 2 million yen		ref	ref	ref
2–4 million yen		1.06 (0.76–1.48)	1.02 (0.73–1.42)	1.02 (0.73–1.42)
4–6 million yen		0.90 (0.60–1.33)	0.83 (0.56–1.24)	0.84 (0.56–1.25)
6–10 million yen		1.29 (0.86–1.95)	1.21 (0.80–1.83)	1.21 (0.80–1.83)
10 million or more		0.76 (0.34–1.69)	0.68 (0.30–1.52)	0.68 (0.31–1.53)
Marital status				
Marriage			ref	ref
No marriage			**0.50 (0.34–0.71)**	**0.49 (0.34–0.71)**
Divorced/Widowed			0.90 (0.58–1.41)	0.90 (0.57–1.40)
Living arrangement				
Living with someone			ref	ref
Living alone			**1.49 (1.05–2.13)**	**1.50 (1.05–2.14)**
Smoking				
Non-smoker			ref	ref
Current smoker			**1.63 (1.29–2.06)**	**1.62 (1.28–2.05)**
Psychological distress				
K6 < 13				ref
K6 ≧ 13				1.23 (0.84–1.80)

Bold items were significant (*p* < 0.05). Model 1: univariable. Model 2: adjusted socioeconomic factors (age, sex, education, job, income,). Model 3: adjusted additional social isolation (living arrangement, marital status) in Model 2 and health factors (smoker, sleeping duration, BMI). Model 4: further adjusted K6 (Kessler Phycological Distress Scale).

## Data Availability

The data that support the findings of this study are available on reasonable request. However, restrictions apply to the availability of these data to protect personal identification; research data are not shared. If any person wishes to verify our data, they are welcome to contact the corresponding author.

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
