# Peer review of "Loneliness and Increased Hazardous Alcohol Use: Data from a Nationwide Internet Survey with 1-Year Follow-Up"

_ijerph, 2022, doi:10.3390/ijerph191912086_

Round 1

Reviewer 1 Report

The research work carried out meets the requirements of soundness and scientific rigour. The introduction is well resolved with updated literature and state of the art.  The methodology used is adequate. The proposed objective is well formulated and answered. The number of participants is relevant. The authors are honest as they have clearly mentioned the 4 limitations of the study: self-reported measures of loneliness ; study restricted the generalisability of the results to the Japanese population in general ; could have led to selection bias and residual confounding. Uses 25 very current references

Author Response

Thank you for your insightful comments. I attached the file that response your comments.

Reviewer 2 Report

This study assessed the association between self-reported loneliness and later drinking in a large, population-based sample. The authors utilized validated self-report instruments, appropriate for this type of study.

I have some comments, mostly minor.

In the abstract and introduction, the authors make several comments pertaining to the COVID-19 pandemic. Yet, no data on the epidemic is reported in the results, so it appears that this is just the broader context in terms of the time frame for this study. Personally, I would prefer to remove references to the COVID-19 pandemic completely, or at the very least move it to the discussion. For instance, how can the authors really conclude that "...Due to the preventive measures aimed at reducing the COVID-19 transmission, people with high loneliness were more susceptible to engaging in hazardous alcohol use." How would we know if this would not have been the case in 2018 as well?

The authors should reference the Kessler-6 instrument.

This is a highly debated topic, but I have become convinced that odds ratios do not represent the outcomes of logistic regression in a meaningful manner. I suggest that the authors present adjusted risk differences or adjusted risk ratios. These can be obtained by running the adjrr command on Stata after running logistic regression. 

The authors state that "We found the relationship between loneliness and drinking may be not linear; low- and medium-risk drinkers may feel less lonely than nondrinkers, probably due to social interaction during drinking. However, alcoholics or severe-risk drinkers have a high possibility to lose social networks due to alcohol-related symptoms and an aggravation of their symptoms due to loneliness (13, 39)." But this study actually assessed the reverse order, that is, whether loneliness at time 1 predicted drinking at time 2.

I know there is research to support the authors' claim that loneliness or lack of social interaction and abstinence may be related. I published one such article myself, but I will not direct the authors to that one, as I do not find it appropriate, but I am certain that there are more than one.

Author Response

Thank you for your insightful comments. I attached our responses to your comments.
